# Multimodal Treatment of Nasopharyngeal Carcinoma in Children, Adolescents and Young Adults-Extended Follow-Up of the NPC-2003-GPOH Study Cohort and Patients of the Interim Cohort

**DOI:** 10.3390/cancers14051261

**Published:** 2022-02-28

**Authors:** Tristan Römer, Sabrina Franzen, Hanna Kravets, Ahmed Farrag, Anna Makowska, Hans Christiansen, Michael J. Eble, Beate Timmermann, Gundula Staatz, Felix M. Mottaghy, Martina Bührlen, Ulrich Hagenah, Alexander Puzik, Pablo Hernáiz Driever, Jeanette Greiner, Norbert Jorch, Stephan Tippelt, Dominik T. Schneider, Gabriele Kropshofer, Tobias R. Overbeck, Holger Christiansen, Triantafyllia Brozou, Gabriele Escherich, Martina Becker, Waltraud Friesenbichler, Tobias Feuchtinger, Wolfram Puppe, Nicole Heussen, Ralf D. Hilgers, Udo Kontny

**Affiliations:** 1Division of Pediatric Hematology, Oncology and Stem Cell Transplantation, Medical Faculty, RWTH Aachen University, 52074 Aachen, Germany; troemer@ukaachen.de (T.R.); sfranzen@ukaachen.de (S.F.); hanna.kravets@rwth-aachen.de (H.K.); afarrag@ukaachen.de (A.F.); amakowska@ukaachen.de (A.M.); 2Pediatric Oncology Department, South Egypt Cancer Institute, Assiut University, Assiut 71515, Egypt; 3Department of Radiotherapy and Radiation Oncology, Hannover Medical School, 30625 Hannover, Germany; christiansen.hans@mh-hannover.de; 4Department of Radiation Oncology, RWTH Aachen University, 52074 Aachen, Germany; meble@ukaachen.de; 5Department of Particle Therapy, University Hospital Essen, West German Proton Therapy Centre Essen (WPE), West German Cancer Centre (WTZ), 45147 Essen, Germany; beate.timmermann@uk-essen.de; 6Section of Pediatric Radiology, University Medical Center Mainz, 55131 Mainz, Germany; gundula.staatz@unimedizin-mainz.de; 7Department of Nuclear Medicine, Medical Faculty, RWTH Aachen University, 52074 Aachen, Germany; fmottaghy@ukaachen.de; 8Department of Radiology and Nuclear Medicine, Maastricht University Medical Center (MUMC+), 6229 Maastricht, The Netherlands; 9Eltern-Kind-Zentrum Prof. Hess, Klinikum Bremen-Mitte, 28211 Bremen, Germany; martina.buehrlen@klinikum-bremen-mitte.de; 10Department of Child and Adolescent Psychiatry, Psychosomatics and Psychotherapy, University Hospital of the RWTH Aachen, 52074 Aachen, Germany; uhagenah@ukaachen.de; 11Division of Pediatric Hematology and Oncology, Department of Pediatrics and Adolescent Medicine, Medical Center, Faculty of Medicine, University of Freiburg, 79106 Freiburg, Germany; alexander.puzik@uniklinik-freiburg.de; 12Department of Pediatric Oncology/Hematology, Charité-Universitätsmedizin Berlin, Humboldt-Universität zu Berlin, 13353 Berlin, Germany; pablo.hernaiz@charite.de; 13Hematology and Oncology Department, Children’s Hospital of Eastern Switzerland, 9006 St. Gallen, Switzerland; jeanette.greiner@kispisg.ch; 14Children Hematology and Oncology, Bethel, 33617 Bielefeld, Germany; norbert.jorch@evkb.de; 15Pediatric Oncology and Hematology, Pediatrics III, University Hospital of Essen, 45147 Essen, Germany; stephan.tippelt@uk-essen.de; 16Clinic of Pediatrics, University Witten/Herdecke, 44137 Dortmund, Germany; dominik.schneider@klinikumdo.de; 17Department of Pediatrics and Adolescent Medicine, Medical University, 6020 Innsbruck, Austria; gabriele.kropshofer@i-med.ac.at; 18Department of Hematology and Medical Oncology, University Medical Center Göttingen, 37075 Göttingen, Germany; tobias.overbeck@med.uni-goettingen.de; 19Department of Pediatric Oncology, Hematology and Hemostaseology, University Hospital Leipzig, 04103 Leipzig, Germany; holger.christiansen@kinderonkologie-leipzig.de; 20Medical Faculty, Department of Pediatric Oncology, Hematology and Clinical Immunology, University Children’s Hospital, Heinrich Heine University, 40225 Düsseldorf, Germany; triantafyllia.brozou@med.uni-duesseldorf.de; 21Department of Pediatric Hematology and Oncology, University Medical Center Hamburg-Eppendorf, 20246 Hamburg, Germany; escherich@uke.de; 22Department for Children and Adolescents, University Hospital Frankfurt, Goethe-University, 60590 Frankfurt am Main, Germany; martina.becker@kgu.de; 23Department of Pediatric Hematology and Oncology, St. Anna Children’s Hospital, Medical University of Vienna, 1090 Vienna, Austria; w.friesenbichler@stanna.at; 24Department of Pediatric Hematology, Oncology and Stem Cell Transplantation, Dr von Hauner University Children’s Hospital, Ludwig Maximilian University, 80337 Munich, Germany; tobias.feuchtinger@med.uni-muenchen.de; 25Institute of Virology, Hannover Medical School, 30625 Hannover, Germany; puppe.wolfram@mh-hannover.de; 26Department of Medical Statistics, RWTH Aachen University Aachen, Pauwelsstrasse 19, 52074 Aachen, Germany; nheussen@ukaachen.de (N.H.); rhilgers@ukaachen.de (R.D.H.); 27Center of Biostatistics and Epidemiology, Sigmund Freud University, Freudplatz 3, 1020 Vienna, Austria

**Keywords:** nasopharyngeal carcinoma, platin-based chemotherapy, radiotherapy, interferon-beta, late toxicities, children, adolescents, young adults

## Abstract

**Simple Summary:**

Multimodal treatment of nasopharyngeal carcinoma (NPC) in children and young adults with induction chemotherapy, followed by radiochemotherapy and interferon-β (IFN-β) maintenance, has been successfully applied in studies NPC-91 and NPC-2003 of the German Society of Pediatric Oncology and Hematology (GPOH). We, here, present updated survival rates of the NPC-2003 study cohort after longer follow-up and include 21 additional patients recruited after closure of the study and treated as per the NPC-2003 study protocol (interim cohort) in our survival analysis. Survival rates remain high after extended follow-up and in the larger cohort with EFS and OS of 94% and 97%, respectively, reinforcing the high antitumor efficacy of this multimodal treatment concept. Seven patients with CR after induction therapy received a reduced radiation dose of 54 Gy, and none of them relapsed. Thus, the reduction of radiation dose seems feasible and has the potential to reduce treatment-related late effects in this vulnerable population.

**Abstract:**

Nasopharyngeal carcinoma (NPC) in children and young adults has been treated within two consecutive prospective trials in Germany, the NPC-91 and the NPC-2003 study of the German Society of Pediatric Oncology and Hematology (GPOH). In these studies, multimodal treatment with induction chemotherapy, followed by radio (chemo)therapy and interferon-beta maintenance, yielded promising survival rates even after adapting total radiation doses to tumor response. The outcome of 45 patients in the NPC-2003 study was reassessed after a median follow-up of 85 months. In addition, we analyzed 21 further patients after closure of the NPC-2003 study, recruited between 2011 and 2017, and treated as per the NPC-2003 study protocol. The EFS and OS of 66 patients with locoregionally advanced NPC were 93.6% and 96.7%, respectively, after a median follow-up of 73 months. Seven patients with CR after induction therapy received a reduced radiation dose of 54 Gy; none relapsed. In young patients with advanced locoregional NPC, excellent long-term survival rates can be achieved by multimodal treatment, including interferon-beta. Radiation doses may be reduced in patients with complete remission after induction chemotherapy and may limit radiogenic late effects.

## 1. Introduction

Nasopharyngeal carcinoma (NPC) is a rare neoplasm in children and adolescents, with a reported incidence of 0.4–0.6 per 1,000,000 persons between 10 and 17 years of age in Germany [1]. It arises due to malignant transformation of EBV-infected nasopharyngeal epithelial cells [2]. Most pediatric patients and young adults with NPC present with advanced locoregional disease, necessitating intensive local and systemic treatment. Successful treatment of NPC in children and adolescents in Germany has been mainly achieved through the implementation of two consecutive prospective trials on behalf of the German Society of Pediatric Oncology and Hematology (GPOH), the NPC-91 and the NPC-2003 study [3,4]. In these studies, multimodal treatment combining induction chemotherapy, followed by concomitant radio (chemo)therapy, and maintenance with interferon-beta (IFN-β), yielded high survival rates. In the NPC-91 study, 59 patients ≤ 25 years of age were included. Of these, 58 patients with NPC stage III or IV received induction chemotherapy with three cycles of methotrexate, cisplatin, and 5-fluorouracil, followed by radiotherapy with a cumulative radiation dose of 59.4 Gy, and postradiation maintenance with IFN-β for six months. After a median follow-up of four years, EFS and OS were 91% and 95%, respectively [3]. However, due to excessive acute toxicity of the three-drug induction regimen with severe mucositis (grade 3/4) in 70% of patients, methotrexate was omitted in the successor trial NPC-2003. Furthermore, in NPC-2003, cisplatin was given concomitantly to radiotherapy, and the radiation dose was reduced to 54 Gy in patients with complete response after induction chemotherapy. The NPC-2003 study comprised 45 patients of ages 8–20 years, of whom 44 patients with stage III or IV disease were treated with the sequence of induction chemotherapy, radiochemotherapy, and adjuvant IFN-β, EFS, and OS were 92% and 97%, respectively. Importantly, none of the five patients who received the reduced radiation dose due to complete response after induction chemotherapy relapsed [4]. The results of these studies led to the publication of treatment recommendations by the GPOH-NPC study group, which have recently been integrated in an international consensus paper for the diagnosis and treatment of NPC in children and adolescents [5,6]. However, with a relatively short median follow-up of 30 months, the reported excellent preliminary results of the NPC-2003 study need to be confirmed by long-term data and higher patient numbers.

The aim of this report is to provide extended follow-up data of the NPC-2003 study cohort to investigate the durability of treatment responses and reassess previously reported survival rates. Furthermore, we include 21 additional patients in this analysis who were treated according to the NPC-2003 study protocol after study closure (called the interim cohort), adding up to a representative cohort of 66 homogeneously treated patients. 

## 2. Materials and Methods

### 2.1. Patients

For the NPC-2003 study, 53 patients from 27 centers in Germany, Austria, and the Netherlands were recruited between 2003 and 2010 [4]. Inclusion criteria were histologically proven NPC and an age ≤ 25 years. Eight patients with major study protocol deviations (*n* = 4), primary distant metastases (*n* = 3), and NPC as secondary malignancy (*n* = 1) were excluded, leaving 45 patients for analysis. Median follow-up of these patients was 85 months (range, 16–168 months), with the last documented examination in January 2019.

After closure of the NPC-2003 study, we continued to be contacted by centers for review of imaging studies and therapy guidance for patients with NPC. In the period from January 2011 until October 2017, when the NPC-GPOH registry 2016 started registration, we were addressed for 30 patients with newly diagnosed NPC. Clinical information on these patients of the so-called interim cohort was provided by the treating physicians. For final analysis, the same inclusion and exclusion criteria were used as for the NPC-2003 study cohort, leading to a cohort of 21 additional patients, treated according to the NPC-2003 study protocol. Patients were excluded from analysis due to primary distant metastases (*n* = 5) or major deviations from the recommended treatment (*n* = 4). The 21 interim patients were from centers in Germany (*n* = 17), Austria (*n* = 2), Switzerland (*n* = 1), and the Netherlands (*n* = 1). Median follow-up of the interim patients was 40 months (range, 3–99 months), with the last documented examination in June 2020.

The study was approved by the Ethics Committee of the Medical Faculty of the RWTH Aachen (EK 189/20).

### 2.2. Diagnosis, Staging, and Response Assessment

Detailed information on diagnostic procedures, treatment protocol, and response assessment in patients of the NPC-2003 study has been provided previously [4]. Histological diagnosis of NPC type II or III according to the modified WHO classification by Krueger and Wustrow was mandatory for inclusion in the NPC-2003 study and interim phase [7]. Local tumor and regional lymph node regions were preferentially visualized by MRI. Recommended staging examinations prior to treatment initiation included PET, chest-CT, and abdominal ultrasound for the detection of distant metastases. It was also recommended to evaluate patients for EBV infection by means of serology, including anti-VCA-IgA, and EBV-PCR.

Tumor stages were defined according to the classification of the International Union against Cancer (UICC) and American Joint Committee of Cancer (AJCC), using the 4th edition for the NPC-2003 patients and the 7th edition for the interim patients [8,9].

For response assessment, MRI, PET, anti-EBV-VCA-IgA, and EBV-PCR were recommended to be repeated after induction chemotherapy, after radiochemotherapy, and after IFN-β maintenance. Response criteria on imaging for all patients were as in the NPC-2003 study, encompassing complete response (CR), very good partial response (VGPR), partial response (PR), stable disease (SD), and progressive disease (PD) [3].

### 2.3. Treatment

In the NPC-2003-GPOH study, patients with tumor stage I or II (T1 N0 M0 or T2 N0 M0) were stratified to the low-risk group and treated with concomitant radiochemotherapy, followed by IFN-β maintenance therapy. Patients with stage III (T3 N0 M0 or T1–3 N1 M0) or IV (T4 N0–3 M0 or T1–4 N2–3 M0) were classified as high-risk patients and treated with three cycles of induction chemotherapy, followed by concomitant radiochemotherapy and IFN-β maintenance. In the interim phase, all patients with stage II or higher were stratified as high-risk and treated according to the treatment protocol of the NPC-2003-GPOH study, as there had been data published in the meantime on an inferior outcome of patients with stage II compared to stage I [10].

Induction chemotherapy consisted of three cycles, each containing cisplatin 100 mg/m^2^ on day 1 and 5-fluorouracil (5-FU) 1000 mg/m^2^/day as continuous infusion on days 1–5. Supportive therapy included leucovorin 25 mg/m^2^, six times every six hours. In the case of ototoxicity CTC grade ≥ 2 or nephrotoxicity with creatinine clearance <50 mL/min/1.73 m^2^, replacement of cisplatin by carboplatin in a dose of 500 mg/m^2^ was recommended. In patients with severe mucositis (grade IV), a reduced 5-FU dose of 1000 mg/m^2^/day over four days was applied in subsequent cycles. Chemotherapy cycles were given at three-week intervals.

Recommended clinical target volume for radiotherapy included the initial primary tumor and all visible macroscopic lymph node metastases (gross tumor volume, GTV) with an anatomically adapted safety margin to cover any subclinical extent, as well as the parapharyngeal and cervical level II lymph nodes in all patients. For patients with stage II–IV disease, cervical lymph nodes of levels III, IV, and V and supraclavicular lymph nodes were also irradiated. The total radiation dose was 45.0 Gy in daily single fractions of 1.8 Gy. In patients having achieved CR as evaluated by MRI and/or PET, a boost of 9.0 Gy to the GTV followed. In cases of incomplete response after induction chemotherapy, a boost of 14.4 Gy was applied. Intensity-modulated radiotherapy (IMRT) was recommended as the modality of choice in order to spare healthy surrounding tissue. Radiotherapy was combined with cisplatin 20 mg/m^2^/day on three consecutive days during the first and last week of irradiation.

After induction chemotherapy and concomitant radiochemotherapy, all patients received IFN-β as maintenance therapy for six months. In the NPC-2003 study, patients mainly received Fiblaferon^®^ (Biosyn, Fellbach Germany) which had been licensed for the treatment of NPC in Germany. Fiblaferon^®^ was given intravenously at a dose of 100,000 IU/kg three days per week, with a maximum single dose of five million IU for patients weighing >50 kg. After 2010, when the production of Fiblaferon^®^ was stopped for non-medical reasons, patients received Rebif^®^ (Merck Europe B.V., Amsterdam, The Netherlands) subcutaneously as off-label treatment. All interim patients therefore received Rebif^®^. The recommended dose was six million IU three times a week.

### 2.4. Late Effects

Late effects were assessed in all patients, in whom at least one documented follow-up examination was reported by the respective treatment center. Late toxicity assessment was done via a questionnaire that was sent out to all participating centers. The questionnaire encompassed 20 items, based on the CTCAE version 4.03, and modified for detecting late effects specific to the treatment of NPC. Grading was done for items ototoxicity, xerostomia, hypothyroidism, hypopitutiarism, retinopathy, and renal failure. For all other items, their presence or absence was recorded, with an open space for comments: secondary malignancy, tinnitus, chronic otitis media, chronic sinusitis, dry eye, soft tissue fibrosis, skin ulceration, cerebral nerve palsy, trismus, pulmonary fibrosis, osteoradionecrosis, cardiac toxicity, dental caries, and cognitive disturbance. Based on the free comments, two additional categories were created: cephalgia and psychological disturbances.

### 2.5. Statistics

The primary objectives were the determination of EFS and OS. EFS was defined as the time from initial diagnosis to first relapse, progression, secondary malignancy, or death from any cause. Patients who remained alive without relapse, progression, or secondary malignancy were censored at the date of last follow-up. For OS, the time from diagnosis to death of any cause was calculated, and patients who remained alive were censored at the last follow-up. EFS and OS were estimated according to the Kaplan–Meier method [11]. Ninety-five percent confidence intervals (CIs) at a given median follow-up time were determined according to the method of Kalbfleisch and Prentice [12]. Response rate to each treatment element and treatment-related late effects were quantitatively described as secondary objectives. Baseline characteristics were described by frequency and percentage or median and range and compared by Fisher’s exact test or *t*-test between NPC-2003 and interim patients. Statistical analyses were performed using the software package SAS (version 9.4; SAS Institute, Inc., Cary, NC, USA). 

## 3. Results

### 3.1. Patient Characteristics

As the NPC-2003-GPOH study cohort had been characterized in detail before [4], we focus here on the interim patients recruited between 2011 and 2017. This cohort included 21 patients without initial distant metastases, 12 of whom were male (57%) and 9 female (43%). The median age at diagnosis was 15 years (range, 11–24 years). One patient had stage II disease (TNM staging T2 N0 M0), and all other patients had stage III or IV disease (T3–4 and/or N2–3). Precise histological classification was stated for 18 patients; 15 were classified as WHO type IIIb and 3 as WHO type IIb. Data on EBV antigen or DNA detection in tumor tissue was available for 17 patients, all of whom were EBV-positive. Results of peripheral blood EBV PCR at the time of diagnosis were available for 15 patients. Of these, 12 had EBV DNA initially detected.

No statistically significant differences were noted in gender distribution, age at diagnosis, histological subtype, and EBV detection rate between the NPC-2003 and interim patients. Distribution of UICC stages differed significantly between the two cohorts. Furthermore, interim patients showed a significantly higher proportion of patients with CR after induction chemotherapy. Follow-up for interim patients was significantly shorter than for patients of the NPC-2003 study, as expected.

Patient characteristics of the two cohorts are summarized in Table 1.

### 3.2. Treatment and Outcome

#### 3.2.1. NPC-2003 Study Cohort

Response to induction chemotherapy, radiochemotherapy, and interferon treatment in the 45 patients of the NPC-2003 study cohort had been described previously [3] and is summarized in Table 1. After a median follow-up of 30 months, three patients relapsed with distant bone metastases six, 10, and 21 months after initial diagnosis, respectively. Two patients received salvage chemotherapy and local irradiation but died of progressive disease 10 and 15 months after relapse diagnosis, respectively. One further patient relapsed with a solitary bone metastasis in the pelvis 21 months after initial diagnosis. He was successfully treated with surgery, salvage chemotherapy, and local irradiation, achieving a second complete remission (Table 2).

With now an extended follow-up of 85 months (range, 16–168 months), no further relapses occurred. Of note, none of the five patients who received a reduced radiation dose of 54.0 Gy due to CR after induction chemotherapy relapsed during extended follow-up (Table 3).

There were two events not related to tumor relapse: a 16-year-old male committed suicide eight months after the end of treatment. Another patient developed protoplasmatic astrocytoma (WHO grade II) of the right temporal lobe four years after initial diagnosis as secondary malignancy. The tumor was completely resected three years after diagnosis, and the patient was reported to be in CR at the last follow-up 13 years after NPC diagnosis. 

#### 3.2.2. Interim Cohort

All patients of the interim phase, including the one patient with stage II disease, were treated according to the NPC-2003 high-risk treatment arm, comprising three cycles of induction chemotherapy, concomitant radiochemotherapy, and subsequent IFN-β maintenance therapy. For evaluation of tumor response, MRI and/or PET were recommended after each treatment element. Definite response status after induction chemotherapy as determined by reference evaluation was available for 17 patients, with documented CR in seven, VGPR in two, and PR in eight patients. No patient was documented to have progressive disease after induction chemotherapy (Table 1).

After IFN-β treatment, seven patients were stated as CR by reference evaluation of MRI and/or PET imaging, and two patients had documented VGPR. For the other patients, reference evaluations were not available (Table 1).

One interim patient relapsed during follow-up. He was a 17-year-old male with stage T4 N2 M0, whose primary tumor was evaluated as PR before and after radiotherapy, and as SD after interferon treatment. Vitality of the residual nasopharyngeal tumor was ruled out by biopsy at the end of treatment. Four months after the end of interferon treatment, PET-CT imaging revealed a bone metastasis in a rib with concomitant parasternal lymph node metastasis. The rib tumor was resected, followed by local irradiation including parasternal lymph node metastasis (39.0 Gy). Another four months later, an axillary lymph node metastasis was histologically diagnosed, and the primary nasopharyngeal tumor slightly progressed in size. With a positive immune reaction for PD-L1 on 70% of tumor cells, he was treated with four cycles of pembrolizumab (200 mg every three weeks). A suspicious lesion in the right pectoral muscle progressed during treatment with pembrolizumab, and biopsy confirmed tumor metastasis. He again received surgery with extensive resection of the right-sided chest wall. Eight months later, imaging studies revealed two new suspicious supra- and infraclavicular lymph nodes in close proximity to the resection area. Biopsy confirmed metastases, with positivity for PD-L1 in 90% of tumor tissue. Repeated therapy with pembrolizumab resulted in complete remission of lymph node metastases after five cycles. However, two new suspicious soft tissue lesions evolved during treatment with pembrolizumab, which was discontinued after a total of nine cycles, and salvage chemotherapy with six cycles of gemcitabine and docetaxel was implemented. The patient eventually achieved remission, with no further relapse for more than two years after omission of treatment.

Characteristics of patients with relapse in the NPC-2003 and interim cohort are summarized in Table 2.

#### 3.2.3. EFS and OS

The EFS and OS rates of the NPC-2003 study cohort were 93.2% (95% CI, 80.3–97.7%) and 95.2% (95% CI, 82.3–98.8%), respectively, after a median follow-up of 85 months (range, 16–168 months; Figure 1).

The EFS and OS of the interim cohort were 94.4% (95% CI, 66.6–99.2%) and 100%, respectively, after a median follow-up of 40 months (range, 3–99 months). When analyzing both cohorts (*n* = 66), EFS, and OS were 93.6% (95% CI, 83.8–97.5%) and 96.7% (95% CI, 87.3–99.2%), respectively, after a median follow-up of 73 months (range, 3–168 months; Figure 2). 

### 3.3. Late Effects

Late effects could be assessed in 41 patients of the NPC-2003 study cohort and 19 patients of the interim phase, in whom at least one documented follow-up examination was reported by the respective treatment center (Figure 3).

The total cohort of 60 patients comprised 40 males and 20 females. Late effects were reported in 49 patients (81.7%), with 34 patients from the NPC-2003 study cohort and 15 interim patients (82.9% and 78.9% of evaluable patients in each cohort, respectively). Twenty-six patients (43.3%) were reported to suffer from two or more late effects, 16 patients from the NPC-2003 study cohort and 10 patients from the interim cohort (39.0% and 52.6% of evaluable patients in each cohort, respectively). 

The most common morbidity was hypothyroidism in 42 patients (70.0%), followed by ototoxicity and xerostomia in 16 (26.7%) and 14 patients (23.3%), respectively. Of the 16 patients with reported ototoxicity, three had severe hearing loss necessitating hearing aids (grade 3) (two patients) or cochlear implants (grade 4) (one patient). All three patients were treated within the NPC-2003 study. Xerostomia was defined as grade 1–2 in all affected patients. Most other reported conditions were also graded as mild, but exact severity grading was not documented in most patients. As aforementioned, one patient of the NPC-2003 study cohort developed astrocytoma (WHO grade II) of the right temporal lobe during extended follow-up, yielding a secondary malignancy rate of 1.7%. One patient in the interim cohort was diagnosed with panhypopituitarism. Of note, there was no reported case of radiotherapy-induced osteonecrosis, and no reported pulmonary or cardiac late effects to date. Comparing the frequency of late effects for the patients of the NPC-2003 study, at 30 months follow-up with the longer follow-up of 85 months, the percentage of patients with hypothyroidism increased from 25% to 78%, and for ototoxicity from 14% to 32%.

## 4. Discussion

We here confirm previously reported survival rates in children and young adults with NPC, treated homogeneously according to the NPC-2003-GPOH study protocol with induction chemotherapy, followed by concomitant radiochemotherapy, and subsequent IFN-β maintenance. The EFS and OS of the NPC-2003 study cohort were not compromised after a median follow-up of seven years, demonstrating durable treatment efficacy and reinforcing our multimodal concept as standard of care for newly diagnosed NPC in children and young adults. Extending the cohort by adding patients of the interim phase also did not affect these results, as only one further patient relapsed, leading to an EFS and OS of 94% and 100%, respectively, after a median follow-up of 40 months in this later cohort. The fact that all four relapses in the total cohort occurred less than two years after initial diagnosis is in accordance with large studies in adults with NPC indicating that later relapses are exceedingly rare after successful initial treatment. However, as all relapses presented as distant bone metastases and only one patient additionally showed secondary progression of the primary tumor, local tumor control seems to be more efficient than control of minimal disseminated disease, leaving room for further optimization of systemic treatment and reduction of radiotherapy for certain patients. 

The preceding NPC-91-GPOH study was the first prospective multicenter trial in the treatment of NPC in children and young adults, evaluating the sequence of induction chemotherapy, followed by radiotherapy, and subsequent IFN-β maintenance for six months [3]. With a DFS and OS of 91% and 95%, respectively, high survival rates were achieved. Therefore, this combination regimen was further pursued in the NPC-2003 study. The introduction of IFN-β in the front-line treatment of NPC was based on its successful use in patients with refractory disease [13,14]. Recent preclinical trials have shown induction of apoptosis in NPC cells via the TRAIL-signaling pathway and activation of tumor-cell eliminating NK cells by IFN-β [15,16]. It is assumed that the immunomodulatory effects of IFN-β may improve systemic disease control in NPC patients, contributing to the high treatment efficacy and low relapse rates in the NPC-GPOH studies. This also has to be interpreted in view of the lower radiation doses applied in the NPC-GPOH studies when compared to other pediatric NPC protocols [3,4,17,18]. The clinical results reported here provide further support for the recommendation of IFN-β treatment, as the much longer follow-up indeed indicates a substantial benefit in long-term tumor control. Of particular concern is the case of a male adolescent in the NPC-2003 cohort who committed suicide eight months after end of IFN-β treatment. He previously suffered from sleep disturbances that first occurred during IFN-β treatment, but definite signs of depression were not reported. Depression and suicidality as side effects of interferon treatment are controversial issues. Some reports on patients with multiple sclerosis hint to an increased rate of depression and suicidal ideation in association with interferon treatment [19,20,21], but most studies failed to confirm this association [22,23,24]. Regarding risk in patients with multiple sclerosis, it is difficult to study the independent effect of a medication on depressive symptoms and suicidality in a disease that per se confers a higher risk for depression [25]. The pre-treatment psychological status is deemed a good parameter for risk assessment, and it is recommended to use IFN-β with caution in patients with pre-existing severe depression or suicidality. For patients treated in the NPC-GPOH studies, there were no further reports of suicidal attempts, but data on psychological well-being were not collected systematically. It cannot be excluded that interferon treatment was an aggravating or even triggering factor for suicide in the patient of the NPC-2003 cohort. Rigorous psychological assessment before, during, and after interferon treatment is mandatory for early recognition of this detrimental side effect.

The main treatment element for local tumor control in NPC is radiotherapy, as surgical resection is not feasible due to the complex anatomical localization and invasiveness of tumors. Radiation doses to the primary tumor site and involved lymph nodes historically vary between 45 and 68 Gy in previous studies in children and adolescents with NPC [3,4,17,18,26,27,28]. In the NPC-91 study, a radiation dose of 59.4 Gy was applied to the primary tumor and 45.0 Gy to locoregional lymph nodes, with only one local relapse during follow-up [3]. Considering this low relapse rate and the high risk for radiogenic late effects, especially in young patients, the local radiation dose was reduced to 54.0 Gy in the NPC-2003 study in those who achieved complete remission after induction chemotherapy [4]. In addition, concomitant cisplatin was implemented due to the results of a beneficial effect of combined radiochemotherapy in adults [29]. It is important to note that none of the five patients of the NPC-2003 cohort who received the reduced radiation dose relapsed during the extended follow-up period. Two further patients of the interim cohort also received a reduced radiation dose of 54.0 Gy, and after 72 and 45 months, respectively, none of them relapsed either. These results suggest that reduction of radiation dose in patients with complete remission is feasible in the context of an effective chemotherapy backbone. With a 98% overall response rate to induction chemotherapy, high chemosensitivity has been demonstrated in the NPC-2003 cohort, despite omission of methotrexate due to excessive toxicity in the NPC-91 study [3,4]. Therefore, high chemosensitivity of pediatric NPC may safely allow reduction of radiation doses, although a prospective randomized approach would be desirable to confirm these observations. Lowering the radiation dose would be an important step toward reduction of toxicity, as radiation-induced late effects are of particular concern in a tumor with excellent primary cure rates, especially in young patients. In addition, the use of IMRT has been reported to cause significantly less severe late toxicities and excellent local tumor control, and is now considered a standard modality for patients with NPC [30,31,32]. Unfortunately, it was not possible to gather sufficient information about the applied radiation technique in most patients. Thus, the contribution of modern recommended radiation techniques, such as IMRT or proton radiation, to reduced treatment-related toxicity and enhanced local tumor control could not be adequately assessed in our series.

The induction chemotherapy used in our study was 5-fluorouracil and cisplatin. Except of the first prospective trial in pediatric NPC by Ghim, reported in 1998, this combination has been used in all other prospective studies of pediatric patients with nasopharyngeal carcinoma, with some studies adding a third agent such as methotrexate or docetaxel [3,4,17,18,26,27,33,34]. Recently, a large randomized trial in adults has shown a significant benefit for the combination of gemcitabine/cisplatin (GP) versus 5-fluoruracil/cisplatin (PF) in patients with NPC and distant metastases or refractory disease [35,36]. However, in adult patients with locoregionally advanced NPC, GP, and PF have so far only been compared in smaller phase II trials, without showing a clear benefit toward one regimen [37,38]. Therefore, in adults, the combination of gemcitabine and cisplatin is recommended by the NCCN and CSCO/ASCO for patients with metastatic disease, whereas for patients with locoregionally advanced NPC, several induction chemotherapy regimens, among them PF and GP, are considered to be of equal value [39,40]. As the OS and EFS/DFS of recent prospective trials in pediatric NPC are high, a change in the chemotherapy backbone in pediatric patients with locoregionally advanced NPC should only be based on a randomized trial, once the best induction regimen in adults has been defined.

Intensification of induction chemotherapy by adding a third drug, such as methotrexate or docetaxel, to cisplatin and 5-FU has not led to superior treatment efficacy in children and young adults, but significantly increased acute toxicity [3,34]. Another approach to increase complete tumor responses after induction chemotherapy, and therefore allow the reduction of radiation dose in more patients, could be the addition of a PD1 checkpoint inhibitor to standard treatment. NPC tumor cells show expression of the immune effector cell inhibiting ligand PD-L1 in about 85% of tumors, providing a biological rationale for blocking the PD-L1/PD1 interaction with specific antibodies [41]. Two phase 2 trials in patients with refractory NPC treated with the anti-PD1 antibodies pembrolizumab or nivolumab have shown an overall response rate of 20% and 26%, and a stable disease rate of 34% and 42%, respectively [42,43]. Here, we also report a patient who was repeatedly treated with pembrolizumab due to refractory disease. This treatment, however, did not result in sustained remission, which could only be achieved by further salvage chemotherapy. Anti-PD1 antibody treatment appears unlikely to induce complete remission as monotherapy. However, combination with chemotherapy could be a means to increase the proportion of patients with complete remission after induction therapy, as this concept has already been proven efficacious in adult patients with advanced non-squamous non-small-cell lung cancer and refractory or metastatic head and neck squamous cell carcinoma [44,45]. For this purpose, a study investigating the effectiveness of the PD1 checkpoint-inhibitor nivolumab, added to the two-drug induction regimen, is underway (EudraCT-Number: 2021-006477-32).

Most surviving NPC patients suffer from late morbidities, with potential impairment of health-related quality of life. Disorders due to the involvement of anatomical structures in proximity to the radiation field, such as hypothyroidism, xerostomia, chronic rhinosinusitis, and dental caries, are among the most commonly reported long-term sequelae. Furthermore, a high rate of ototoxicity, and less frequently, secondary malignancies, both probably as a consequence of the combined toxic effect of platin-based chemotherapy and locoregional radiotherapy, are of special concern [46]. In the combined NPC-2003 and interim cohort, more than 80% of patients were reported to suffer from at least one significant morbidity, with hypothyroidism, hearing impairment, and xerostomia being the most common. Cheuk et al. reported a similar proportion of patients with late effects in 59 children with NPC, but with a different distribution regarding the frequency of specific morbidities. Of note, sensorineural hearing loss was more common in his study, with more than 50% of patients affected, while the incidence of hypothyroidism was higher in our cohort (70% vs. 56%). One could speculate that the reported lower rate of hearing impairment in our cohort was due to the reduced radiation dose in combination with the relatively low dose of cisplatin during radiochemotherapy. It was shown that the cumulative incidence of primary hypothyroidism depends on the radiation dose applied to locoregional lymph nodes, with a significantly elevated risk with doses of more than 50 Gy [46]. As all patients in our cohort except one received radiation doses of more than 50 Gy, including the ones who received a reduced dose, prevention of hypothyroidism is perhaps not achievable in these dose ranges that are necessary for sufficient local tumor control. This is in line with the fact that three out of eight patients who received a reduced radiation dose were also diagnosed with hypothyroidism (Table 3). When comparing the late effects in patients of the GPOH studies with those observed in the aforementioned retrospective study from the St. Jude Children’s Research Hospital (38105 Memphis, TN, USA), despite a similar high rate of overall late effects, a lower incidence of severe morbidities was noted in the GPOH studies. We believe this to be mostly due to the lower radiation dose compared to all other prospective pediatric NPC trials. Furthermore, the overall low rate of severe systemic late effects, such as nephrotoxicity or cardiopulmonary complications, indicates an acceptable toxicity profile of the two-drug chemotherapy regimen with cisplatin and 5-fluorouracil. Nevertheless, when comparing the rates of the two most relevant late toxicities, hypothyroidism, and ototoxicity, between the former report by Bührlen et al. and our current report, it is obvious that morbidity is increasing with longer follow-up. Bührlen et al. reported hypothyroidism and ototoxicity in 25% and 14% of patients, respectively, after a median follow-up of 30 months, while we described proportions of 78% and 32%, respectively, after a median follow-up of 85 months. Cumulation over time points toward the fact that these morbidities are pathogenetically based on a persistent toxic effect rather than a one-hit damaging mechanism. Reduction of frequency and severity of treatment-related long-term morbidities in NPC survivors should be a major attempt in future trials, including a special focus on psychological and quality-of-life aspects. Such trials should encompass the validation of biomarkers on outcomes such as EBV-DNA, or circulating tumor DNA, as well as tumor genetics, in order to better stratify patients into risk groups. Improved risk group stratification could not only allow dose reductions in radiotherapy, but also reduction of induction chemotherapy or its substitution by immmunotherapy in patients with a favorable risk profile. Further insights into the processes involved in tumorigenesis with a special focus on the interplay between EBV infection and host immunity will surely pave the way toward a more biologically based treatment with high efficacy and less long-term toxicity in patients with NPC.

Statistical comparison of the NPC-2003 study and interim cohort revealed a significant difference in the distribution of tumor stages according to the UICC/AJCC staging system (Table 1). This may be due to the fact that two different editions of the UICC/AJCC cancer staging manual were used for the NPC-2003 and interim patients. The main difference between the 4th edition (used for the NPC-2003 study patients) and the 7th edition (used for the interim patients) of the UICC/AJCC staging system is a downstaging trend in the 7th edition (see [6,7] for detailed comparison). This may explain the significantly higher proportion of stage III and significantly lower proportion of stage IV patients in the interim cohort. 

Another observation is the significant difference in complete response rates after induction chemotherapy and radiochemotherapy between the NPC-2003 and interim cohort (11% vs. 41%, *p* < 0.05, and 45% vs. 73%, *p* = 0.051, respectively). A possible explanation could be the higher rate of lower stage disease in the interim cohort, leading to a better or more rapid tumor response in these patients. However, the above-mentioned differences in the classification systems used for disease staging, with a downstaging trend in the more recent staging system used for the interim patients, and the overall low number of patients for whom reference response evaluation was available, prevents meaningful conclusions regarding differences in response rates between the cohorts.

The combined CR and VGPR rates after the end of treatment in the NPC-2003 and interim patients, for whom reference evaluations were available, were 95% and 100%, respectively. The proportion of patients with CR or VGPR after induction chemotherapy and radiochemotherapy was 39% and 79%, respectively, in the NPC-2003 cohort, and 53% and 82%, respectively, in the interim cohort. One could argue that tumor responses appear to be rather slow in NPC, with a relatively high proportion of significant residual lesions after induction chemotherapy and radiochemotherapy, which is in contrast to the exceptionally high local control rate during long-term follow-up. It must be kept in mind that only locoregionally advanced diseases with significant extension of tumor lesions without any attempts at surgical resection are included in this analysis, which could explain the perceived slow responses in imaging studies. The phenomenon of residual lesions in MRI is often seen in NPC, not necessarily representing residual vital tumor tissue. A better means to visualize residual vital tumors is PET-CT, which, however, can also lead to false-positive findings due to unspecific or inflammatory tracer uptake in the nasopharyngeal space. As most patients have not received both MRI and PET-CT after each treatment element, meaningful correlation between findings is hampered, and many false-positive results may contribute to the high proportion of residual lesions during the treatment course. However, close follow-up with regular imaging studies in patients with residual lesions is mandatory, and biopsy to rule out vital tumors is advised in patients with any signs of progression.

Finally, it has to be kept in mind that patients with initial distant metastases were excluded from our analysis, as the treatment and prognosis of these patients are highly variable. With five patients in the NPC-91 study and three patients in the NPC-2003 study, primary metastatic disease is a rare scenario.

## 5. Conclusions

By analyzing longer follow-up data of the NPC-2003-GPOH study and expanding the study cohort by 21 equally treated interim patients, we add further support to our multimodal treatment concept with cisplatin-based induction chemotherapy, combined radiochemotherapy, and subsequent IFN-β maintenance as standard of care for children and young adults with newly diagnosed NPC. It appears that further reduction of radiation dose is feasible in patients with complete response to induction chemotherapy without losing antitumor efficacy, as none of seven patients who received a radiation dose of 54 Gy relapsed during follow-up. With EFS and OS of 94% and 97%, respectively, exceptionally high survival rates have now been achieved, and future studies should focus on further attempts to prevent treatment-related long-term morbidities in the most vulnerable group of the young.

## Figures and Tables

**Figure 1 cancers-14-01261-f001:**
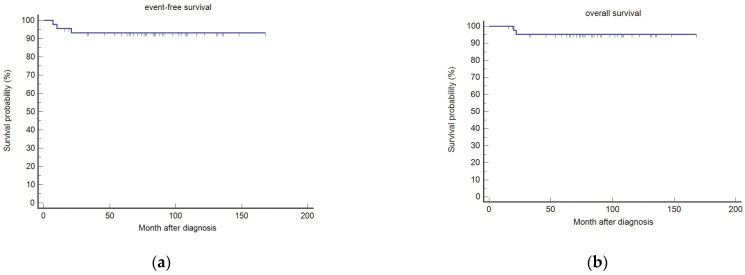
Event-free (**a**) and overall survival (**b**) for patients of the NPC-2003 study (*n* = 45) after the extended median follow-up of 85 months.

**Figure 2 cancers-14-01261-f002:**
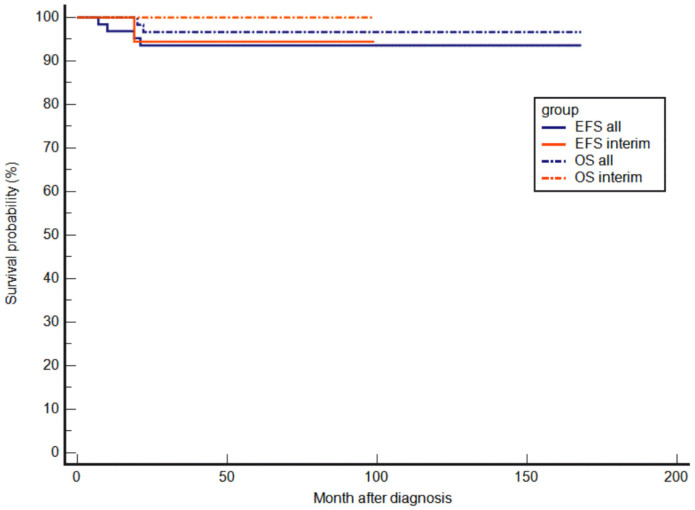
Event-free and overall survival for all patients of the NPC-2003 study and the interim phase (*n* = 66) after a median follow-up of 73 months, and for the interim patients alone (*n* = 21) after a median follow-up of 40 months.

**Figure 3 cancers-14-01261-f003:**
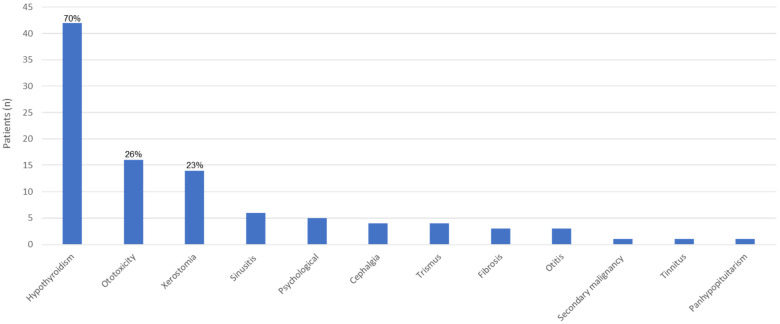
Distribution of late effects in 60 patients of the NPC-2003 study and interim cohort.

**Table 1 cancers-14-01261-t001:** Patient characteristics of the NPC-2003 study and interim cohort.

Characteristics	All Patients (*n* = 66)	NPC-2003 Study Patients (*n* = 45)	Interim Patients (*n* = 21)	*p* Value ^f^
Gender–no. (%)				0.4112
Female	23 (34.8)	14 (31.1)	9 (42.9)	
Male	43 (65.2)	31 (68.9)	12 (57.1)	
Age at diagnosis–year				0.3119
Median	15	15	15	
Range	8–24	8–20	11–24	
UICC stage ^a^–no. (%)				<0.001
I	1 (1.5)	1 (2.2)	0	
II	1 (1.5)	0	1 (4.8)	
III	14 (21.2)	4 (8.9)	10 (47.6)	
IV	50 (75.8)	40 (88.9)	10 (47.6)	
Histological type ^b^–no. (%)				0.2556
I (squamous cell)	0	0	0	
II (non-keratinizing)	6 (9.5)	3 (6.7)	3 (16.7)	
III (undifferentiated)	57 (90.5)	42 (93.3)	15 (83.3)	
EBV detection in tumor tissue (antigen or DNA) ^c^–no. (%)				0.152
Positive	45 (90)	28 (84.8)	17 (100)	
Negative	5 (10)	5 (15.2)	0	
Response to induction chemotherapy–no. (%) ^d,e^				<0.05
CR	12 (20)	5 (11.4)	7 (41.2)	
VGPR	14 (23.3)	12 (27.3)	2 (11.8)	
PR	33 (55)	26 (59)	8 (47)	
SD	1 (1.7)	1 (2.3)	0	
PD	0	0	0	
Response after radiochemotherapy–no. (%) ^d^				0.051
CR	25 (51)	17 (44.7)	8 (72.7)	
VGPR	13 (26.5)	13 (34.2)	1 (9.1)	
PR	11 (22.5)	8 (21.1)	2 (18.2)	
SD	0	0	0	
PD	0	0	0	
Response after end of interferon treatment–no. (%) ^d^				0.8992
CR	32 (69.6)	25 (67.6)	7 (77.8)	
VGPR	12 (26.1)	10 (27)	2 (22.2)	
PR	0	0	0	
SD	0	0	0	
PD	2 (4.3)	2 (5.4)	0	
Follow up–mo				<0.05
Median	73	85	40	
Range	3–168	16–168	3–99	
Relapse–no. (%)	4 (6.1)	3 (6.7)	1 (4.8)	0.1573

^a^ For the NPC-2003 study, tumor staging was performed according to the 4th edition of the UICC/AJCC Cancer Staging Manual from 1993. For the interim patients, the 7th edition from 2011 was used. ^b^ Histological type was determined according to the WHO classification modified by Krüger and Wustrow [7]. Three patients of the interim cohort with no detailed information on histological subtype were excluded. ^c^ Sixteen patients with no information about EBV detection in tumor tissue were excluded (12 NPC-2003 study patients, four interim patients). ^d^ Response status after induction chemotherapy, radiochemotherapy, and interferon treatment was determined by reference evaluation of MRI (G.S.) and/or PET-CT (F.M.M.). Patients with no documented reference evaluation were excluded: four interim patients for response to neoadjuvant chemotherapy, seven NPC-2003 study and 10 interim patients for response after radiochemotherapy, and eight NPC-2003 study and 12 interim patients for response after the end of interferon treatment. ^e^ For assessment of response to induction chemotherapy, the one low-risk patient in the NPC-2003 study who did not receive neoadjuvant chemotherapy was excluded from this analysis. This patient (UICC stage I) achieved PR five weeks after radiochemotherapy and CR two months after starting interferon treatment. ^f^ Statistical comparisons between the NPC-2003 study and interim patients were done by *t*-test for age and follow-up, and by Fisher’s exact test for all other variables.

**Table 2 cancers-14-01261-t002:** Characteristics of patients with relapse.

No.	Cohort	Sex	Age at Initial Diagnosis–Year	UICC Stage	Histol-ogical Type	Total Radiation Dose to Primary Tumor and Locoregional Lymph Nodes–Gy	Response at End of Primary Treatment	Time of Relapse (After Initial Diagnosis)–Month	Relapse Site	Treatment and Outcome
1	NPC-2003	m	11	IV	III	59.4	PR	6 (after RT, before IFN treatment)	Distant: bone	Multiple bone metastases; salvage chemotherapy, RT; died of PD 15 months after relapse diagnosis
2	NPC-2003	m	15	IV	III	59.4	PR	10 (4 months after start of IFN treatment)	Distant: bone, pleura	Salvage chemotherapy, RT; died of PD 10 months after relapse diagnosis
3	NPC-2003	m	19	IV	III	59.4	CR	21 (9 months after end of IFN treatment)	Distant: bone	Second CR after surgery, salvage chemotherapy and RT; alive in CR at follow-up 72 months
4	Interim	m	17	IV	III	69.8	PR	17 (4 months after end of IFN treatment)	Distant: bone	Multiple relapses, extensive surgery, RT, repeated treatment with pembrolizumab, remission achieved after salvage chemotherapy with gemcitabine and docetaxel; alive in remission at follow-up 75 months

**Table 3 cancers-14-01261-t003:** Characteristics of patients who received a reduced radiation dose due to CR after induction chemotherapy.

No.	Cohort	Sex	Age at Initial Diagnosis–Year	UICC Stage	Histological Type	Documented Radiation Dose–Gy	Follow-Up–Month	Status at Last Follow-Up
1	NPC-2003	m	12	IV	n.s.	54.0	131	Alive in CR; learning disability, obesitas, hypothyroidism
2	NPC-2003	m	12	IV	III	54.0	122	Alive in CR; chronic dental/parodontal disease
3	NPC-2003	m	12	III	III	54.0	108	Alive in CR; no further information
4	NPC-2003	m	16	IV	III	54.0	76	Alive in CR; hypothyroidism, chronic tinnitus
5	NPC-2003	m	13	IV	III	54.0	34	Alive in CR; hypothyroidism, high-frequency hearing deficit
6	Interim	f	19	III	II	54.0	72	Alive in CR; restriction of mouth opening
7	Interim	m	14	III	II	54.0	45	Alive in CR; chronic sinusitis
8 ^a^	Interim	m	11	III	III	43.2	17	Alive in CR; hearing deficit, xerostomia

^a^ This patient received a reduced radiation dose of 43.2 Gy due to discontinuation of RT because of an ischemic brain insult. He was evaluated as CR after induction chemotherapy by local MRI, but no reference evaluation was done. n.s. = not stated.

## Data Availability

Datasets supporting the results of this work are available to editors, referees, and readers promptly upon request.

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
