# Peer review of "Multimodal Treatment of Nasopharyngeal Carcinoma in Children, Adolescents and Young Adults-Extended Follow-Up of the NPC-2003-GPOH Study Cohort and Patients of the Interim Cohort"

_cancers, 2022, doi:10.3390/cancers14051261_

Round 1

Reviewer 1 Report

The authors present long term results of an interesting trial/cohort of pediatric patients with nasopharynx cancer. 

Results: What instrument was used for late toxicity assessment? More details about precise grades of toxicity are expected from a prospective trial and would have been more desirable. 

Discussion: It is worth discussing the role of gemcitabince/cisplatin induction therapy as this approach has been used successfully for adult patients. Also, given the high EFS and OS, reduction of adjuvant systemic treatment (INF beta) also merits discussion. Were EBV DNA titers collected? Correlation of outcome with this biomarker is also of interest and should be included in the discussion. 

Conclusion: This should focus on (1) high EFS/OS achieved (2) high locoregional control with reduced doses of radiation especially after CR to induction. The remainder of the conclusion should ideally be part of the discussion.

Reviewer 2 Report

I think this a very interesting and also quite important paper. The authors show long-term results of multimodal treatment of NPC in children, young adults and adolescents. The paper is well written

minor comments:

  • IN the introduction it would be helpful if the evolution of treatment from NPC-91 to NPC-2003 would be briefly explained.
  • IN table 1 the response rates to neoadjuvant chemo and radiochemo are significantly different between NPC-2003 and the interim cohort. THis should at least be mentioned in the discussion
  • Also the response rate to radiochemo. The complete response rate in NPC-2003 was only 44.7 %. How can this be, since basically all tumors were controlled. Even after interferon treatment only 67.6% were complete responders. Was tumor regression so slow that after many months the tumor was still visible on imaging?
  • In the legend to Table 1 and reference Krüger and Wustrow has the wrong number. It is reference 6, not 5.
  • Itwould be interesting to see the difference in toxicity between 30 months of follow-up  in the original NPC-2003 publication and 85% of follow up in the current analysis. THis could be shown in a table.
